# Reliability in long-term clinical studies of disease-modifying therapies for relapsing-remitting multiple sclerosis: A systematic review

Rosa C. Lucchetta[1]*, Letícia P. Leonart[1], Marcus V. M. Gonçalves[2], Jefferson Becker[3], Roberto Pontarolo[1], Fernando Fernandez-Llimós[4], Astrid Wiens[1]*

1 Graduate Program in Pharmaceutical Sciences, Federal University of Paraná, Curitiba, Brazil,
2 Universidade da região de Joinville (UNIVILLE), Joinville, Brazil, 3 Brain Institute and School of Medicine, Pontifical Catholic University of Rio Grande do Sul, Porto Alegre, Brazil, 4 Laboratory of Pharmacology, Department of Drug Sciences, Faculty of Pharmacy, University of Porto, Porto, Portugal

* rc.lucch@yahoo.com.br (RCL); astrid@ufpr.br (AW)

## Abstract

### Background

Although relapsing-remitting multiple sclerosis (RRMS) has a chronic course, little information is known about the comparison between the disease-modifying therapies (DMT) for long-term outcomes. We aimed to conduct a systematic review of randomized clinical trial (RCT) extension and observational studies to examine the efficacy and safety of all available DMT for RRMS, compare the evidence with that derived from mid-term studies, and investigate whether the published long-term data are robust and reliable enough to inform clinical decision-making concerning RRMS treatment.

### Method

PubMed, Scopus, and manual searches were performed until October 2019. The clinical outcomes of long- and mid-term studies were compared. ROBINS-I was used to assess the methodological qualities of the long-term studies. PROSPERO number CRD42019123361.

### Results

Nineteen long-term studies (9,018 participants) were included in the systematic review. All studies presented serious or critical risks of bias that were mainly due to confounding, selection, and missing data biases. The annualised relapse rates (ARR) observed in the long-term studies are lower (better) than those from the mid-term studies for most treatments. The main reason for this ARR decrease could be a selection bias for good responders in the long-term studies, since many studies show a loss of patients between the mid- and long-term phases. The safety profiles depend on the study, follow-up, report, and outcome (i.e., discontinuation or number of patients with at least one serious adverse event).

**Data Availability Statement:** The data can be accessed via the following repository:

https://osf.io/kge7f/?view_only=
8f45544cc8924d8781a1b783588c6052.

**Funding:** This study was funded by the Institutional Development Support Programme of the National Health System (Proadi-SUS) and Hospital Alemão Oswaldo Cruz (grant number 01/2017). The funders had no role in study design, data collection and analysis, decision to publish, or preparation of the manuscript.

**Competing interests:** RCL reports personal fees from Biogen and Roche; JB reports grants and personal fees from Biogen, Novartis, Roche and Teva and personal fees from Bayer, Ipsen, Merck Serono, Sanofi, outside the submitted work. LPL, MVMG, RP, FFL and AW declare that they have no conflict of interest. This does not alter our adherence to PLOS ONE policies on sharing data and materials. These institution as well as funder had no role in any of the phases of the study (i.e., study design, data collection, data analysis, interpretation, writing of the report and responsibility for submission).

## Conclusion

The currently available long-term data for patients with RRMS exhibit serious or critical risks of bias that preclude robust comparisons between long-term studies. High quality comparative observational studies with long-term follow-ups or RCT extensions with intention-to-treat analyses are needed to support clinical and regulatory practice. Until reliable long-term evidence is available, neurologists should continue to base their conduct on mid-term studies, patient's experience and, most importantly, patient's needs and predictor factors, according to personalized medicine.

## Introduction

Multiple sclerosis is a debilitating chronic inflammatory disease that affects the central nervous system [1]. Relapsing-remitting multiple sclerosis (RRMS) is the most common type of multiple sclerosis (85% of cases) [2] and is characterised by relapses, i.e., the appearance of new symptoms or exacerbations of previous ones, followed by a period of full or partial recovery without new symptoms and progression [3].

Disease-modifying therapies (DMT) are used to improve the course of RRMS and reduce the severity of symptoms [4]. The evidence-based efficacies of all internationally approved DMT are well described in systematic reviews with network meta-analyses (NMAs) that show the superiority of alemtuzumab, natalizumab, and ocrelizumab for limiting the annualised relapse rate (ARR) compared with other DMT when considering a median follow-up time of two years [5–7]. However, RCTs usually have a limited duration and fail to assess long-term outcomes, which are important given the chronicity of RRMS. Other types of studies, such as observational comparative cohort studies and RCT extensions, should be considered for guiding treatment decisions. These long-term studies are methodologically poorer than RCTs, but when properly conducted, they are an important source of information about long-term safety and sustained efficacy. A recent NMA of the short- and long-term clinical outcomes of patients with clinically isolated syndrome identified that the risk of developing clinically definite multiple sclerosis was reduced after early DMT treatment compared with delayed DMT [8]. However, this reduction was not identified in another systematic review of long-term RRMS treatments.

Therefore, we aimed to perform a systematic review of studies reporting efficacy and safety outcomes of long-term DMT use for RRMS, compare the evidence with that derived from mid-term studies (previously published RCTs), and investigate if the published long-term data from cohort and RCTs studies are robust and reliable enough to inform clinical decision-making in RRMS.

## Materials and methods

The systematic review was performed in accordance with the Preferred Reporting Items for Systematic Reviews and Meta-Analyses (PRISMA) [9] (S1 Table in S1 Appendix) and Cochrane Collaboration recommendations [10], and it was registered in the International Prospective Register of Systematic Reviews (PROSPERO) with the number CRD42019123361.

Electronic searches were conducted in the PubMed and Scopus databases without any time limit or language restriction (until October 2019). Trial registration databases (ClinicalTrials.

gov) and the reference lists of reviews and included studies were also searched. Complete search strategies are provided in S2 Table in S1 Appendix.

We included studies that fulfilled the following inclusion criteria according to the PICOS acronym:

## Population

Patients aged 18 years and older diagnosed with RRMS; studies evaluating RRMS with other forms of multiple sclerosis (i. e. clinically isolated syndrome, primary progressive multiple sclerosis or secondary progressive multiple sclerosis) were excluded.

## Intervention and control

DMT used as monotherapy (dose comparisons and head-to-head studies against placebo or no treatment), including ALE12/ ALE24: alemtuzumab, 12 or 24 mg/ day per 5 days and 12 months later per 3 days; BG240BID/ TID: dimethyl fumarate, 240 mg, twice-times daily or three-times daily; FING0.5QD/ 1.25QD: fingolimod, 0.5 or 1.25 mg daily; GA20QD: glatiramer acetate, 20 mg daily; IFNA22TIW/ IFNA44TIW: interferon 1a beta 22 or 44 µg three-times weekly; IFNB250EOD: interferon 1b beta, 250 µg, every other day; IFNA30QW: interferon 1a beta, 30 µg weekly; PLA: placebo.

## Outcomes

Annualised relapse rate (ARR, which is the primary outcome of most of the mid-term studies [6]), discontinuation due to adverse events (DAE), and the number of patients with at least one serious adverse event (SAE).

## Studies

Prospective, or retrospective comparative cohort studies, randomised phase II or later controlled trials (including post-hoc analyses), and multi- or single-arm extensions of RCTs with at least 36 months of follow-up. Equivalence studies were excluded.

For studies that evaluated a switch in therapy, we included only the arms with at least 36 months of continuing follow-up. Studies that considered at least one of the aforementioned outcomes were included.

Two researchers independently screened the titles and abstracts of the retrieved studies to identify irrelevant records. In a second stage, full-text articles were also independently evaluated by two researchers according to the inclusion and exclusion criteria. Discrepancies were reconciled in consensus meetings using a third researcher as a referee.

The following data were independently extracted by two researchers: (i) study characteristics (authors' names, year of publication, trial design, sample size, evaluated DMT, mean follow-up, diagnostic criteria, and sponsor); (ii) baseline data (patients' sex and age, disease duration, or symptoms onset); and (iii) clinical outcomes.

The baseline data and clinical outcomes of the long-term studies ($\geq$ 36 months) were compared with those from mid-term studies ($>$ 3 and $<$ 36 months) that were recovered from a recently published systematic review [6]. The data were tabulated according to the ARR and standard deviation; when a study reported the confidence interval, it was converted to a standard deviation. The DAE and SAE were reported according to the number of patients with the outcome, sample size, and percentage.

The critical evaluations of the risks of bias of the studies were conducted by two independent reviewers using the Risk of Bias in Non-randomised Studies of Interventions (ROBINS-I)

tool [11]. In the absence of consensus, points of disagreement were resolved by the opinion of a third researcher. The risks of bias of the mid-term studies were assessed using the Cochrane Collaboration revised Risk of Bias assessment tool [12], and the results have been published in a previous systematic review [6].

## Results

Our systematic review identified 1,760 records in the electronic databases after duplicate removal and obtained two by manual search. Of these, 1,699 were considered irrelevant during the screening, and 38 were excluded during the full-text appraisal (Fig 1 and S3 Table in S1 Appendix). The remaining 25 records (19 studies) comprised 14 RCT, and five observational studies and were included in the qualitative synthesis (S4 Table and S7 Table in S1 Appendix). The articles were published between 2003 and 2018. In total, 9,018 participants (median: 147; interquartile range: 83–249) were included, and 5,468 (60%) were women (three studies did not mention the proportion of patients' genders). Five studies (26%) evaluated a switch in therapy. Altogether, 14 dosages of DMT were identified, six (32%) studies compared active therapies (head-to-head), seven (37%) compared doses, five (26%) were non-comparative, and one (5%) evaluated the active treatment against no treatment. No studies assessing natalizumab, ocrelizumab, or teriflunomide fulfilled the inclusion criteria.

A qualitative comparison of the mid- and long-term baseline data revealed they were very similar, except ATTAIN that included a population with relapses within the previous 2 years

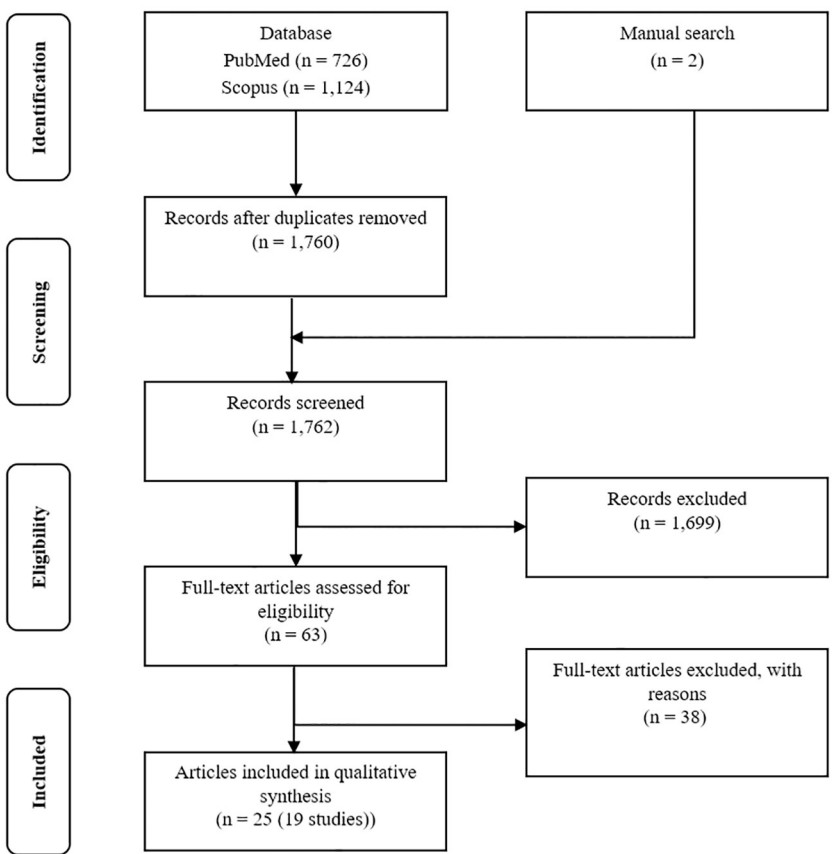

**Fig 1. Study selection.** The reasons are showed in supporting information.

of 0.36 and 0.45 in contrast with other studies which report 2.3 or even more. The studies had follow-up times ranging from 3 to 8.5 years. The main characteristics of the studies are presented in Table 1 (additional characteristics are presented in S4 Table in S1 Appendix).

Additionally, 28 mid-term RCTs were considered for comparison, and their characteristics have been previously reported [6]. In summary, the mid-term articles were published between 1995 and 2018 with a median of 2011. Most of the studies included both treatment-naïve and treatment-experienced patients 12 (40%) or did not report this information 11 (38%), 6 (20%) included only treatment-naïve participants, and 1 (3%) assessed only treatment-experienced patients. Most of the studies had a follow-up of 2 years (median 2; interquartile range: 1–2).

The methodological qualities of the long-term studies are presented in S5 Table in S1 Appendix. All studies were found to have serious or critical methodological problems. The non-comparative RCT extension studies were all deemed to have critical risks of bias because the lack of a comparison group automatically precludes the comparability of such a study to an RCT (the gold standard), and the ROBINS-I questions assess comparability between groups, whether concerning baseline characteristics or concerning patient follow-up. All comparative RCT extensions and cohort studies presented with serious risks of bias, and the following domains were primarily responsible for these classifications: 'bias due to confounding factors', 'selection bias', and 'missing data bias'. Most of the studies did not report any attempt to control key confounders (e.g., adjusting the analyses), which limits the comparability between arms. Most studies only included the patients who tolerated the drug and did not discontinue the treatment during the core study into the extension phase. Many studies also lacked missing data management, which varied between 0% and 83% of the dropout rate.

The methodological qualities of the included mid-term studies were recently published [6]. In summary, most of the studies presented a 'low risk of bias' (58%), which was followed by 'some concerns' (25%). The domain that most frequently scored as a 'high risk of bias' was the measurement of the outcome (due to the lack of the masking of the assessors).

PIFN125Q2W, ALE12, ALE24 (unapproved dose), and CLA3.5 were the DMT with lower ARRs followed by BG240BID, BG240TID (unapproved dose), FING0.5QD, FING1.25QD (unapproved dose), GA20QD, GA40TIW, IFNA30QW, IFNB250EOD, IFNA22TIW, and IFNA44TIW. Comparison of the long- and mid-term results revealed that the ARR of the long-term studies was lower than the ARR of the mid-term studies. However, this finding lacked statistical analysis support (Table 2).

The safety scenario was less consistent; the different safety profiles depended on the study, outcome evaluated (discontinuation or the number of patients with at least one serious adverse event), follow-up time, and outcome measure or report. The annual incidences of DAE and SAE were reported by 5 and 2 long-term studies, respectively, and the numbers of patients who presented with an event of DAE and SAE in the complete follow-up were reported by 8 and 9 long-term studies, respectively. The proportions of events were similar between the different treatment studies, but ALE12 and FING1.25QD (unapproved dose) exhibited reduced DAE from the mid-term to the long-term endpoints. Regarding SAE, CLA3.5 reported an increased proportion from the mid-term to the long-term (S6 Table in S1 Appendix).

## Discussion

We investigated the long-term effects of DMT in RRMS through a systematic review of 19 studies (9,018 participants). Recent NMAs of DMT in RRMS [5, 7, 13] have been limited to RCTs that have reported only short- (< 3 months) and mid-term outcomes (> 3 and < 36 months). In our study, we aimed to more comprehensively summarise the clinical outcomes of DMT by expanding the follow-up to fully capture the comparative effect of long-term

**Table 1. Characteristics of the studies included in the systematic review.**

| Study | Year | Type of study | Follow-up (months) | Evaluated alternatives | N Participants (n Women) | Age, mean in years (SD) | Baseline EDSS, mean (SD) | Disease duration, mean in years (SD) |
|---|---|---|---|---|---|---|---|---|
| ADVANCE/ ATTAIN | 2018 | RCText | 48 | PIFN125Q2W | 376 (271) | 39.0 (9.7) | 2.4 (1.3) | 8.5 (6.3) [a] |
| | | | | PFIN125Q4W | 354 (251) | 38.1 (9.9) | 2.4 (1.4) | 8.1 (6.1) [a] |
| CARE-MS I | 2017 | RCText | 60 | ALE12 | 335 (NR) | NR | NR | NR |
| CARE-MS II | 2017 | RCText | 60 | ALE12 | 393 (NR) | NR | NR | NR |
| CAMMS223 | 2008 | RCT | 36 | IFNA44TIW | 111 (71) | 32.8 (8.8) | 1.9 (0.8) | 1.4 (0.2; 6.3) [ab] |
| | | | | ALE12 | 112 (72) | 31.9 (8.0) | 1.9 (0.7) | 1.3 (0.1; 3.5) [ab] |
| | | | | ALE24 | 110 (71) | 32.2 (8.8) | 2.0 (0.7) | 1.2 (0.3; 3.2) [ab] |
| | 2012 | RCText | 80 | IFNA44TIW | 47 (30) | 33.1 (8.4) | 2.0 (0.7) | 1.4 (0.2; 3.1) [ab] |
| | | | | ALE12 and ALE24 | 151 (98) | 31.8 (8.7) | 1.9 (0.8) | 1.4 (0.1; 3.3) [ab] |
| CLARITY | 2017 | RCText | 48 | CLA3.5 | 186 (124) | 40.6 (10.5) | 2.5 (0.0; 6.5) [b] | 10.4 (7.1) |
| CMSSG | 1998 | RCText | 36 | GA20QD | 99 (NR) | 34.7 (5.8) | 2.8 (1.2) | 7.4 (4.9) |
| | 2000 | RCText | 72 | GA20QD | 101 (72) | 37.5 (5.8) | 2.7 (1.6) | NR |
| | 2005 | RCText | 96 | GA20QD | 142 (NR) | NR | NR | NR |
| | 2006 | RCText | 120 | GA20QD | 232 (170) | 35.5 (6.4) | NR | 8.3 (5.1) |
| CombiRx | 2013 | RCT | 36 | IFNA30QW | 250 (173) | 37.6 (10.2) | 2.0 (1.2) | 1.4 (4.0) |
| | | | | GA20QD | 259 (185) | 39.0 (9.5) | 1.9 (1.2) | 1.0 (2.9) |
| ENDORSE | 2016 | RCText | 60 | BG240BID→BG240BID | 501 (352) | 39.7 (9.1) | 2.5 (1.3) | 6.9 (5.0); 10.0 (6.5) [a] |
| | | | | BG240TID→BG240TID | 502 (354) | 40.0 (9.1) | 2.4 (1.1) | 6.4 (4.9); 9.3 (6.1) [a] |
| | | | | PLA→BG240BID | 249 (178) | 39.9 (8.8) | 2.5 (1.1) | 6.8 (5.3); 10.1 (6.7) [a] |
| | | | | PLA→BG240TID | 248 (166) | 40.5 (9.4) | 2.5 (1.2) | 7.0 (5.4); 9.5 (6.2) [a] |
| | | | | GA20QD→BG240BID | 118 (86) | 38.2 (8.5) | 2.6 (1.2) | 6.2 (5.0); 9.0 (5.8) [a] |
| | | | | GA20QD→BG240TID | 118 (76) | 39.5 (9.5) | 2.7 (1.2) | 6.3 (4.8); 9.2 (6.3) [a] |
| FREEDOMS | 2015 | RCText | 48 | PLA→FING0.5QD | 155 (106) | 38.1 (8.3) | 2.4 (1.3) | 7.8 (5.9) [a] |
| | | | | PLA→FING1.25QD | 145 (107) | 36.6 (9.2) | 2.4 (1.2) | 8.4 (7.1) [a] |
| | | | | FING0.5QD→FING0.5QD | 331 (234) | 36.5 (8.6) | 2.3 (1.3) | 8.0 (6.6) [a] |
| | | | | FING1.25QD→FING1.25QD | 289 (204) | 37.2 (8.9) | 2.4 (1.3) | 8.2 (6.7) [a] |
| GALA | 2017 | RCText | 36 | GA20QD | 943 (641) | 37.4 (9.4) | 2.8 (1.2) | 7.7 (6.7) [a] |
| OWIMS | 2005 | RCT | 36 | IFNA22TIW→IFNA22TIW | 95 (NR) | NR | NR | NR |
| | | | | IFNA44TIW→IFNA44TIW | 98 (NR) | NR | NR | NR |
| | | | | PLA→IFNA22TIW | 49 (NR) | NR | NR | NR |
| | | | | PLA→IFNA44TIW | 51 (NR) | NR | NR | NR |
| PRISMS | 2005 | RCText | 48 | PLA→IFNA22TIW | 189 (127) | 34.8 (29.3; 39.8) [c] | 2.5 (1.2) | 5.4 (3.0; 11.2) [c] |
| | | | | PLA→IFNA44TIW | 184 (121) | 35.6 (28.4; 41.0) [c] | 2.5 (1.3) | 6.4 (2.9; 10.3) [c] |
| | | | | PLA→IFNA22TIW | 85 (62) | 35.8 (NR) | 3.0 (2.5) | NR |
| Saida | 2017 | RCText | 36 | FING0.5QD | 47 (33) | 34.9 (9.0) | 2.4 (1.9) | 8.2 (6.6) [a] |
| | | | | FING1.25QD | 46 (31) | 35.7 (8.8) | 1.9 (1.7) | 7.6 (5.5) [a] |
| TRANSFORMS | 2015 | RCText | 54 | FING0.5QD | 356 (235) | 36.5 (8.7) | 2.2 (1.3) | 7.3 (6.2) [a] |
| Moccia, 2018 | 2018 | Obs | 102 [d] | IFNA44TIW | 191 (123) | 31.4 (8.3) | 1.5 (1.0; 3.5) [b] | 2.7 (2.8) [a] |
| | | | | IFNA30QW | 168 (104) | 32.3 (7.8) | 1.5 (1.0; 3.5) [b] | 2.8 (2.7) [a] |
| | | | | IFNB250EOD | 148 (93) | 34.2 (8.5) | 1.5 (1.0; 3.5) [b] | 2.5 (2.7) [a] |
| Onesti, 2003 | 2003 | Obs | 36 | IFNB250EOD | 83 (53) | 33.3 (7.5) | 1.9 (1.0) | 8.2 (6.6) |
| | | | | No treatment | 83 (53) | 33.6 (7.6) | 1.9 (1.1) | 7.3 (6.6) |
| Patti, 2006 | 2006 | Obs | 72 | IFNA30QW | 62 (36) | 36.8 (7.3) | NR | 5.8 (6.0) |
| | | | | IFNB250EOD | 64 (38) | 36.6 (7.7) | NR | 5.9 (6.3) |

*(Continued)*

**Table 1.** (Continued)

| Study | Year | Type of study | Follow-up (months) | Evaluated alternatives | N Participants (n Women) | Age, mean in years (SD) | Baseline EDSS, mean (SD) | Disease duration, mean in years (SD) |
|---|---|---|---|---|---|---|---|---|
| **Río, 2005** | 2005 | Obs | 60 | IFNB250EOD | 152 (99) | 33.2 (9.4) | 2.4 (1.1) | 6.1 (5.2) |
| | | | | IFNA44TIW | 127 (90) | 35.3 (9.3) | 2 (0.9) | 6.1 (5.8) |
| | | | | IFNA30QW | 103 (76) | 31.3 (9.1) | 2 (1.1) | 5.1 (4.9) |
| **Ruggieri, 2003** | 2003 | Obs | 60 | IFNB250EOD | 56 (32) | 37 (21–52) | NR | NR |
| | | | | IFNA30QW | 38 (24) | 34 (19–50) | NR | NR |
| | | | | IFNA22TIW | 18 (12) | 36 (19–48) | NR | NR |
| | | | | IFNB250EOD→IFNA30QW | 10 (6) | 40 (21–50) | NR | NR |

[a] Symptom onset;

[b] Median (range);

[c] Median (interquartile range); → switch therapy. SD: standard deviation; EDSS: Expanded Disability Status Score; RCText: randomized clinical trial extension; Obs: observational study; NR: not reported. ALE12/ ALE24: alemtuzumab, 12 or 24 mg/ day per 5 days and 12 months later per 3 days; BG240BID/ TID: dimethyl fumarate, 240 mg, twice-times daily or three-times daily; FING0.5QD/ 1.25QD: fingolimod, 0.5 or 1.25 mg daily; GA20QD: glatiramer acetate, 20 mg daily; IFNA22TIW/ IFNA44TIW: interferon 1a beta 22 or 44 μg three-times weekly; IFNB250EOD: interferon 1b beta, 250 μg, every other day; IFNA30QW: interferon 1a beta, 30 μg weekly; PLA: placebo.

studies and demonstrated their limited value for supporting clinical decision-making and practice guidelines.

The comparison of mid-term RCTs (i.e., the gold-standard) with long-term RCT extensions and observational studies (i.e., real-world data) aims to identify potential differences in outcomes that could be explained by population differences. Although it would be useful to have strong evidence about the long-term outcomes of DMT, our findings highlight the importance of being cautious when considering RCT extensions and observational studies to support clinical practice because of their important limitations that can compromise the validity of their evidence. Despite these limitations, some multiple sclerosis treatment guidelines usually consider evidence extracted from mid- and long-term studies, including extension studies, to support their recommendations [14]. Although MS neurologists expert base their conduct on the patient's experience or personalized medicine (i.e. patient's needs and predictor factors) [15, 16], neurologists not expert in MS have a limited evidence to facilitate making decision, considering both clinical trials, observational studies and guidelines.

Thus far, there is no consensus regarding whether an RCT extension is an observational or an interventional study. The literature exhibits a tendency to classify these types of study as observational [17, 18] because they do not start a new therapy, and more importantly, because, except CLARITY Extension [19], no appropriate randomisation exists at the beginning of the extension phase. Randomisation and masking are essential characteristics that guarantee the superiority of RCTs, but they are lost during an extension phase [18, 20, 21]. Thus, we decided to evaluate both cohort and RCT extension studies using the ROBINS-I tool in our systematic review. Our position is in agreement with the FREEDOMS researchers who registered a RCT extension as an observational study in ClinicalTrials.gov [22]. Unfortunately, other RCT extension studies that were included in our systematic review were registered as interventional or only mentioned the same NCT from an original RCT [19, 23–29]. Notably, even if RCT extensions were considered interventional studies, their methodological qualities, as assessed with a tool for RCT assessment, would result in a high risk of bias classification due to the lack of randomisation, awareness of the therapy by the assessors, missing data domains, and even because comparability is lost when only one arm is followed. The number of extension studies

**Table 2. Comparison between mid- and long-term annualised relapse rate.**

| | Study | Mid-term (Only RCT) | | Long-term (RCT, Extension and observational) | |
|---|---|---|---|---|---|
| | | 3- to 12-month | 24-month | 36 to 48-month | ≥ 60-month |
| | | ARR (SD) [n] | | ARR (SD) [n–% of patients from the original study] | |
| ALE12 | CARE-MS I | - | 0.18 (0.49) [376] [a] | - | 0.16 (NR) [349–93%] |
| | CARE-MS II | - | 0.26 (0.63) [426] [a] | - | 0.21 (NR) [357–84%] |
| | CAMMS223 | - | - | 0.11 (0.22) [112–100%] | 0.12 (0.19) [112–100%] [a] |
| | CAMMS223 | - | - | - | 0.11 (0.19) [112–100%] [a] |
| ALE24 | CAMMS223 | - | - | 0.08 (0.19) [110–100%] | 0.11 (0.16) [110–100%] [a] |
| | CAMMS223 | - | - | - | 0.13 (0.19) [110–100%] [a] |
| BG240BID | CONFIRM | - | 0.22 (NR) [359] | - | - |
| | DEFINE | - | 0.17 (0.36) [410] | - | - |
| | ENDORSE [b] | - | - | 0.14 (NR) [501–65%] | 0.14 (NR) [442–57%] |
| | ENDORSE [b] | - | - | 0.14 (NR) [468–61%] | - |
| | ENDORSE [b] | - | - | 0.11 (NR) [192–43%] [d] | - |
| | ENDORSE [b] | - | - | 0.12 (NR) [84–19%] [c] | - |
| BG240TID | CONFIRM | - | 0.20 (NR) [345] | - | - |
| | DEFINE | - | 0.19 (0.42) [416] [a] | - | - |
| | ENDORSE [b] | - | - | 0.16 (NR) [502–68%] | 0.17 (NR) [428–58%] |
| | ENDORSE [b] | - | - | 0.20 (NR) [461–62%] | - |
| | ENDORSE [b] | - | - | 0.16 (NR) [188–43%] [d] | - |
| | ENDORSE [b] | - | - | 0.12 (NR) [76–17%] [c] | - |
| CLA3.5 | CLARITY | - | 0.14 (0.27) [433] [a] | 0.10 (0.24) [186–43%] [a] | - |
| FING0.5QD | Saida 2017 | 0.50 (1.12) [57] [b] | | 0.25 (NR) [57–100%] | - |
| | FREEDOMS | - | 0.18 (0.37) [425] [a] | 0.19 (0.32) [425–100%] [a] | - |
| | FREEDOMS II | - | 0.21 (0.39) [358] [a] | - | - |
| | TRANSFORMS | 0.16 (0.48) [429] [a] | - | 0.17 (NR) [243–57%] | 0.16 (NR) [243–57%] |
| FING1.25QD | Saida 2017 | 0.41 (1.03) [54] [a] | - | 0.21 (NR) [54–100%] | - |
| | FREEDOMS | - | 0.16 (0.32) [429] [a] | 0.16 (0.32) [429–100%] [a] | - |
| | FREEDOMS II | - | 0.20 [370] (0.39) [a] | - | - |
| | TRANSFORMS | 0.20 (0.52) [420] [a] | - | - | - |
| GA20QD | COMBIRX | - | - | 0.11 (NR) [359–100%] | - |
| | ECGA | 0.81 (NR) [119] | - | - | - |
| | CORAL | 0.33 (0.81) [586] | - | - | - |
| | BEYOND | - | 0.34 (NR) [448] | - | - |
| | Calabrese 2012 | - | 0.50 (0.40) [48] | - | - |
| | CMSSG | - | 0.59 (NR) [125] | 1.34 (1.52) [99–79%] [a] | 0.42 (0.44) [101–81%] [a] |
| | CMSSG | - | - | - | 0.20 (NR) [142–57%] [a e] |
| | CONFIRM | - | 0.29 (NR) [350] | - | - |
| | REGARD | - | 0.29 (NR) [378] | - | - |
| | GATE | - | 0.40 (1.77) [357] [d] | - | - |
| GA40TIW | GALA | 0.33 (0.78) [943] [a] | | 0.21 (NR) [716–76%] | - |
| IFNA30QW | EVIDENCE | 0.65 (NR) [338] | - | - | - |
| | TRANSFORMS | 0.33 (0.85) [431] [a] | - | - | - |
| | Calabrese 2012 | - | 0.50 (0.60) [47] | - | - |
| | MSCRG | - | 0.67 (NR) [158] | - | - |
| | BRAVO | - | 0.26 (0.02) [447] | - | - |
| | INCOMIN | - | 0.70 (0.90) [92] | - | - |
| | COMBIRX | - | - | 0.16 (NR) [250–100%] | - |
| | Río 2005 (Ob) | - | - | 0.24 (0.51) [89–100%] | 0.27 (0.56) [37–42%] |
| | Río 2005 (Ob) | - | - | 0.29 (0.60) [63–71%] | - |
| | Patti 2006 (Ob) | - | - | 0.61 (NR) [62–100%] | 0.35 (NR) [62–100%] |
| | Patti 2006 (Ob) | - | - | 0.55 (NR) [62–100%] | 0.32 (NR) [62–100%] |
| | Moccia 2018 (Ob) | - | - | - | 0.35 (0.43) [168–100%] |

(*Continued*)

**Table 2.** (Continued)

| | Study | Mid-term (Only RCT) | | Long-term (RCT, Extension and observational) | |
| --- | --- | --- | --- | --- | --- |
| | | 3- to 12-month | 24-month | 36 to 48-month | ≥ 60-month |
| | | ARR (SD) [n] | | ARR (SD) [n–% of patients from the original study] | |
| **IFNA22TIW** | DMSG | - | 0.70 (NR) [143] | - | |
| | PRISMS | - | 1.82 (NR) [189] | 0.80 (NR) [167–88%] | - |
| | OWIMS | - | - | 0.83 (NR) [95–100%] | - |
| **IFNA44TIW** | EVIDENCE | 0.54 (NR) [339] | - | - | - |
| | Kappos 2011 | 0.36 (0.71) [54] [a] | - | - | - |
| | TENERE | 0.22 (0.81) [104] [a] | - | - | - |
| | Calabrese 2012 | - | 0.40 (0.60) [46] | - | - |
| | CARE-MS I | - | 0.39 (1.15) [187] [a] | - | - |
| | CARE-MS II | - | 0.52 (0.91) [202] [a] | - | - |
| | OPERA I | - | 0.29 (0.62) [411] [a] | - | - |
| | OPERA II | - | 0.29 (0.68) [418] [a] | - | - |
| | REGARD | - | 0.30 (NR) [386] | - | - |
| | CAMMS223 | - | - | 0.36 (0.40) [111–100%] [a] | 0.35 (0.35) [111–100%] [a] |
| | CAMMS223 | - | - | - | 0.35 (0.35) [111–100%] [a] |
| | Río 2005 (Ob) | - | - | 0.32 (0.62) [62–100%] | 0.41 (0.80) [17–27%] |
| | Río 2005 (Ob) | - | - | 0.41 (0.72) [46–74%] | - |
| | OWIMS (Ob) | - | - | 0.77 (NR) [98–100%] | - |
| | Moccia 2018 (Ob) | - | - | - | 0.32 (0.59) [191–100%] |
| | PRISMS | - | 1.73 [184] (NR) | 0.72 (NR) [167–91%] | - |
| **IFNB250EOD** | DMSG | - | 0.71 (0.67) [158] [a] | - | - |
| | INCOMIN | - | 0.50 (0.70) [96] [a] | - | - |
| | BEYOND | - | 0.36 (NR) [897] | - | - |
| | Río 2005 (Ob) | - | - | 0.35 (0.61) [134–100%] | 0.24 (0.48) [114–85%] |
| | Río 2005 (Ob) | - | - | 0.30 (0.67) [127–95%] | - |
| | Onesti 2003 (Ob) | - | - | 0.40 (NR) [83–100%] | - |
| | Patti 2006 (Ob) | - | - | 0.50 (NR) [64–100%] | 0.45 (NR) [64–100%] |
| | Patti 2006 (Ob) | - | - | 0.55 (NR) [64–100%] | 0.41 (NR) [64–100%] |
| | Moccia 2018 (Ob) | - | - | - | 0.34 (0.47) [148–100%] |
| **PIFN125Q2W** | ADVANCE | 0.23 (NR) [438] | 0.18 (NR) [437] | 0.20 (NR) [375–86%] | 0.06 (NR) [185–34%] |
| **PIFN125Q4W** | ADVANCE | 0.29 (NR) [438] | 0.29 (NR) [438] | 0.27 (NR) [354–81%] | 0.12 (NR) [170–39%] |
| | ADVANCE | - | - | 0.20 (NR) [322–74%] | - |
| **No treatment** | Onesti 2003 (Ob) | - | - | 0.40 (NR) [83–100%] | - |

[a]: given as confidence interval and converted to standard deviation;

[b]: ENDORSE = CONFIRM + DEFINE Extension;

[c]: switch therapy (GA → BG240) with ≥ 3-year in BG240;

[d]: switch therapy (placebo → BG240) with ≥ 3-year in BG240;

[e]: PLA and GA in original study (n = 251); ALE12/ ALE24: alemtuzumab, 12 or 24 mg/ day per 5 days and 12 months later per 3 days; BG240BID/ TID: dimethyl fumarate, 240 mg, twice-times daily or three-times daily; FING0.5QD/ 1.25QD: fingolimod, 0.5 or 1.25 mg daily; GA20QD: glatiramer acetate, 20 mg daily; IFNA22TIW/ IFNA44TIW: interferon 1a beta 22 or 44 µg three-times weekly; IFNB250EOD: interferon 1b beta, 250 µg, every other day; IFNA30QW: interferon 1a beta, 30 µg weekly; PLA: placebo; Ob: observational; NR: not reported.

has increased in the last decade despite the lack of standardisation of their methodological qualities, which compromises their reliabilities to inform clinical practice. The loss of randomisation is a special concern in long-term studies because the patients who enter the extension phase belong to a selected group that could tolerate [20] and positively respond to the therapy during the original RCT [21]. ATTAIN study is a good example since it is reported a frequency of relapse within the last 2 years of 0.36 and 0.45 for PIFN125Q2W and PIFN125Q4W groups,

respectively, which is very below than ARR reported by other DMTs. The confounding bias domain in the ROBINS-I assesses how a study deals with a lack of randomisation by adjusting for potential confounders, which is rarely performed. In observational cohort studies, adjusting for potential confounders is more frequent, but this was not the case in the majority of the studies included in our systematic review.

Another concern due to the observational design or the extension of the clinical trial is the absence or loss of blinding patients and assessors. In the case of RRMS, the absence of blinding can be critical, since the main clinical efficacy outcomes are related to relapse, which is a subjective result, considering the range of different definitions for relapse. For example, some authors define that the relapse must last at least 24 hours [30], others 48 hours [31]; some authors define that relapse should increase $\geq 1$ point in two scores of functional systems (FSS) or $\geq 2$ points in an FSS [32], while others define relapse should increase $\geq 1$ in the score of the Expanded Disability Status Scale (EDSS) if the previous EDSS score was $\leq 5.5$ and $\geq 0.5$ if the previous EDSS score was $\geq 6$ [33]. Thus, these discrepancies between the definitions show how relapse can be considered a subjective outcome and, therefore, the patient or assessor awareness of the therapy can influence the assessment, contributing to different ARR results between mid- and long-term studies for the same DMT.

The lack of adjustment for covariates—as an observational study must guarantee, blinding and maintenance of randomization—as an experimental study must guarantee, may be some reasons for lower (i.e., better) ARRs reported in several of the long-term studies compared with their mid-term predecessors. For example, PRISMA presented ARRs of 1.82 for the mid-term studies and 0.83 in the long-term phase. The main reason for this unexpected decrease could be a selection bias for good responders in the long-term studies after 12% of the patients were lost between the mid- and long-term phases. In our systematic review, a quarter of the studies had a dropout rate above 20% before the beginning of the extension phase. Hemming et al. proposed the use of intention-to-treat analysis with respect to the baseline group of patients entering into a RCT; i.e., they should be treated as a responder or non-responder depending on the reason for not continuing in the extension study [34].

Another potential reason for discrepancies in ARR reported for the same DMT among several studies is the lack of a common adjustment: while some studies adjust the ARR for EDSS [35], others consider age [36], sex or still an unadjusted analysis [37]. Additionally, data can be modelled by negative binomial regression [38] and others by Poisson regression [39].

Comparing mid- and long-term results across different studies is further compromised due to the variability in the starting point (i.e., mid-term). For example, IFNA22TIW presented an ARR that ranged from 0.70 in DMSG study to 1.82 in PRISMS study, and IFNA44TIW presented an ARR lower than 0.55 for all studies; however, PRISMS reported an ARR of 1.73. This important variability in efficacy in the mid-term studies can be explained by several differences in the conduction of these studies: DMSG is a study with high risk of bias, while PRISMS is a study with low risk of bias; PRISMS is an old study that used Poser's 1983 diagnostic criteria, whereas most studies assessing IFNA44TIW used the McDonald criteria (2001 to 2010), which might have resulted in different characteristics of the included patients when the more sensitive diagnostic criteria that allow for earlier diagnosis were used [40]. Differences in the proportions of patients with highly active or rapidly evolving severe conditions in the mid- and long-term studies could also explain these discrepancies, but most of the mid- and long-term studies did not report this information, since the terms highly active or rapidly evolving severe RRMS have been more used only in the last decade [41, 42].

Differences in the risk of bias and population can also limit the comparability between long-term studies. For example, CARE-MS II ($\geq 60$ months) included only treatment-experienced

patients and reported an ARR for ALE12 of 0.21, while CAMMS223 ($\geq$ 60-month) included only treatment-naïve patients and reported an ARR for ALE12 of 0.12.

We identified a consistently higher proportion of patients with at least one SAE in the long-term compared with the mid-term studies. However, this result could be misleading because most studies did not define SAE and could consider multiple sclerosis relapses as a SAE instead of a therapeutic failure [43]. Another issue precluding the comparison between studies is the inconsistent manner of reporting safety outcomes. Some studies reported the annual incidence, while others reported the proportion of patients with a presenting event during the complete follow-up period. For example, CARE-MS II reported the number of patients who discontinued ALE12 each year together with the number of patients who continued ALE12 therapy each year. However, CAMMS223 reported only that five patients discontinued due to adverse events over the five years of study. For CAMMS223, the calculation of the incidence of adverse events rate is not possible. The only studies that reported safety outcomes as incidence per patient-year were CARE-MS I and II, and GALA.

One limitation of our study, as with any systematic search, is that missing studies could exist. However, a grey literature search found no additional studies, and only one additional study was found through the manual searches. These findings reinforce the quality of our search. We were unable to perform meta-analyses of the long-term outcomes because of the poor reporting of these outcomes in primary studies of DMT in RRMS.

In conclusion, the current available evidence regarding long-term safety and efficacy outcomes cannot sufficiently contribute to clinical decision making in patients with advanced RRMS because the studies have critical or serious risks of bias due to the inclusion of a selected population composed of good responders in both efficacy and safety. The conduction of high quality comparative observational studies with long-term follow-ups or RCT extensions with intention-to-treat analyses is needed to support clinical and regulatory practice. Until reliable long-term evidence is available, neurologists should continue to base their conduct on mid-term studies, patient's experience in terms of effectiveness and safety and, most importantly, patient's needs and predictor factors, according to personalized medicine.

## Supporting information

**S1 Appendix.**
(DOC)

## Author Contributions

**Conceptualization:** Rosa C. Lucchetta, Fernando Fernandez-Llimós.

**Data curation:** Rosa C. Lucchetta.

**Formal analysis:** Rosa C. Lucchetta.

**Funding acquisition:** Rosa C. Lucchetta, Astrid Wiens.

**Investigation:** Rosa C. Lucchetta, Letícia P. Leonart.

**Methodology:** Rosa C. Lucchetta.

**Project administration:** Rosa C. Lucchetta.

**Resources:** Rosa C. Lucchetta.

**Software:** Rosa C. Lucchetta.

**Supervision:** Fernando Fernandez-Llimós, Astrid Wiens.

**Validation:** Letícia P. Leonart.

**Visualization:** Fernando Fernandez-Llimós, Astrid Wiens.

**Writing – original draft:** Rosa C. Lucchetta.

**Writing – review & editing:** Rosa C. Lucchetta, Letícia P. Leonart, Marcus V. M. Gonçalves, Jefferson Becker, Roberto Pontarolo, Fernando Fernandez-Llimós, Astrid Wiens.

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
