## [Decision Letter · Decision Letter 0]

16 Jan 2020

PONE-D-19-31331

Reliability in long-term clinical studies of disease-modifying therapies for relapsing-remitting multiple sclerosis: A systematic review

PLOS ONE

Dear Dr. Lucchetta,

Thank you for submitting your manuscript to PLOS ONE. After careful consideration, we feel that it has merit but does not fully meet PLOS ONE’s publication criteria as it currently stands. Therefore, we invite you to submit a revised version of the manuscript that addresses the points raised during the review process.

Both Reviewer's suggested that the manuscript should be more clinically oriented, appealing to the general neurology audience with no expertise in trial design or statistics.

We would appreciate receiving your revised manuscript by Mar 01 2020 11:59PM. To enhance the reproducibility of your results, we recommend that if applicable you deposit your laboratory protocols in protocols.io, where a protocol can be assigned its own identifier (DOI) such that it can be cited independently in the future. For instructions see: http://journals.plos.org/plosone/s/submission-guidelines#loc-laboratory-protocols

We look forward to receiving your revised manuscript.

Kind regards,

Aristeidis H. Katsanos, MD, PhD

Academic Editor

PLOS ONE

Journal Requirements:

"I have read the journal's policy and the authors of this manuscript have the following competing interests: RL reports personal fees from Biogen and Roche; JB reports grants and personal fees from Biogen, Novartis, Roche and Teva and personal fees from Bayer, Ipsen, Merck Serono, Sanofi, outside the submitted work. LL, MG, RP, FFL and AW declare that they have no conflict of interest."

Reviewers' comments:

Reviewer's Responses to Questions

**Comments to the Author**

1. Is the manuscript technically sound, and do the data support the conclusions?

Reviewer #1: Yes

Reviewer #2: Yes

2. Has the statistical analysis been performed appropriately and rigorously? 

Reviewer #1: Yes

Reviewer #2: Yes

3. Have the authors made all data underlying the findings in their manuscript fully available?

Reviewer #1: Yes

Reviewer #2: Yes

4. Is the manuscript presented in an intelligible fashion and written in standard English?

Reviewer #1: Yes

Reviewer #2: Yes

5. Review Comments to the Author

Reviewer #1: This a well written metanalysis of long term open label DMT studies in MS.The statistical analysis is more than adequate.The authors conclude that these studies have low quality and clinicians should not rely on them for clinical decisions. Some points to be addressed :

In the abstract, the authors state that "peginterferon, alemtuzumab, and cladribine are the most effective DMT considering ARR.:" Since all these studies have numerous biases and since some DMTs (natalizumab, ocrelizumab and teriflounomide) were not studied in this metanalysis, this statement may be misleading to the readers and would better be avoided. In addition , it is not clearly supported in the main body of the manuscript. In fact , authors rely on tables and do not fully discuss their findings in the main text.

Lines 172-175 : DMTs and doses studied are presented in abbreviations that are not described in text, One has to search to the tables in order to clarify which abbreviation is which DMT. The authors should rephrase these sentences in a way that readers understand the sabstances mentioned.

In general, the authors empasize on findings of the programs used in this metanalysis. This makes the manuscript sound quite " technical" and not clinician friendly . Some recommendations for clinicians could be also added in the text.

Reviewer #2: Congratulations for your systematic review and the meaningful approach to a hot issue, such as clinical management of MS patients. I think that your work confirmed the fact that mid-term as well as long-term studies should be evaluated with criticism from the clinician perspective, because of the risk of bias that is so high lighted in your paper, though all of those "limitations" in comparing studies are known barriers in clinical decision making. Thus, multiple sclerosis treatment guidelines, support their recommendations based on evidence from RCTs and long term studies, as you mentioned in your paper, but they are just recommendations providing some guidance to non experts neurologists in MS, they do not substitute clinical judgement nor provide personalised medicine practice.

My overall impression of the paper was that it was more focused to the difficulties of achieving the aims of the study than the report of efficacy and safety outcomes. It was clear enough to me that the published longterm data are not reliable enough to inform clinical decision making, but my estimation is, that is something that is already known in the clinical setting. I would prefer the paper to be more clinical friendly-oriented and provide some light in the chaotic MS landscape.

6. PLOS authors have the option to publish the peer review history of their article (what does this mean?). If published, this will include your full peer review and any attached files.

Reviewer #1: No

Reviewer #2: No

---

## [Author Response · Author response to Decision Letter 0]

2 Mar 2020

Dear Aristeidis H. Katsanos,

We would like to thank you and the reviewers for the constructive review and for the suggestions for improving our manuscript for publication in Plos One.

We send below the responses and comments to the editor and reviewers:

Editor: Thank you for submitting your manuscript to PLOS ONE. After careful consideration, we feel that it has merit but does not fully meet PLOS ONE’s publication criteria as it currently stands. Therefore, we invite you to submit a revised version of the manuscript that addresses the points raised during the review process.

Both Reviewer's suggested that the manuscript should be more clinically oriented, appealing to the general neurology audience with no expertise in trial design or statistics.

Authors: We appreciate the consideration and analyze the discussion to add more clinical implications and, in addition, we add a clear recommendation to the clinician. In addition, we address other concerns highlighted by the reviewers (all changes are highlighted in blue in the text). Finally, both in the text and in the system, we have corrected institutional information about one of the authors and information about competing interests, as follows: "RL reports personal fees from Biogen and Roche; JB reports grants and personal fees from Biogen, Novartis, Roche and Teva and personal fees from Bayer, Ipsen, Merck Serono, Sanofi, outside of the submitted work. LL, MG, RP, FFL and AW declare that they have no conflict of interest. This does not alter our adherence to PLOS ONE policies on sharing data and materials. This funder had no role in any of the study phases (ie study design, data collection, data analysis, interpretation, report writing and submission responsibility) ".

Comments to the Author

5. Review Comments to the Author

Reviewer #1: This a well written metanalysis of long term open label DMT studies in MS.The statistical analysis is more than adequate. The authors conclude that these studies have low quality and clinicians should not rely on them for clinical decisions. Some points to be addressed : In the abstract, the authors state that "peginterferon, alemtuzumab, and cladribine are the most effective DMT considering ARR.:" Since all these studies have numerous biases and since some DMTs (natalizumab, ocrelizumab and teriflounomide) were not studied in this metanalysis, this statement may be misleading to the readers and would better be avoided. In addition , it is not clearly supported in the main body of the manuscript. In fact , authors rely on tables and do not fully discuss their findings in the main text.

Authors: We fully agree with the reviewer and the abstract has been revised (excerpts from the abstract are highlighted).

Reviewer #1: Lines 172-175 : DMTs and doses studied are presented in abbreviations that are not described in text, One has to search to the tables in order to clarify which abbreviation is which DMT. The authors should rephrase these sentences in a way that readers understand the sabstances mentioned.

Authors: We fixed it (excerpts from the methods are highlighted).

Reviewer #1: In general, the authors empasize on findings of the programs used in this metanalysis. This makes the manuscript sound quite " technical" and not clinician friendly . Some recommendations for clinicians could be also added in the text.

Authors: We agree. The discussion has been revised and the changes are highlighted.

Reviewer #2: Congratulations for your systematic review and the meaningful approach to a hot issue, such as clinical management of MS patients. I think that your work confirmed the fact that mid-term as well as long-term studies should be evaluated with criticism from the clinician perspective, because of the risk of bias that is so high lighted in your paper, though all of those "limitations" in comparing studies are known barriers in clinical decision making. Thus, multiple sclerosis treatment guidelines, support their recommendations based on evidence from RCTs and long term studies, as you mentioned in your paper, but they are just recommendations providing some guidance to non experts neurologists in MS, they do not substitute clinical judgement nor provide personalised medicine practice.

Authors: We fully agree with the reviewer and add that consideration to the personalized medicine in the discussion and conclusion.

Reviewer #2: My overall impression of the paper was that it was more focused to the difficulties of achieving the aims of the study than the report of efficacy and safety outcomes. It was clear enough to me that the published longterm data are not reliable enough to inform clinical decision making, but my estimation is, that is something that is already known in the clinical setting. I would prefer the paper to be more clinical friendly-oriented and provide some light in the chaotic MS landscape.

Authors: In fact, concluding about the efficacy and safety of therapies considering the high methodological limitation of the studies is neither possible nor recommended, with the risk of suggesting that clinicians make their decisions based on the clinical findings identified here. On the other hand, we agree that some light should be given to clarify to clinicians the practical implications of these limitations. Therefore, the discussion was revised to incorporate these aspects and practical recommendations (highlighted excerpts).

---

## [Decision Letter · Decision Letter 1]

18 Mar 2020

PONE-D-19-31331R1

Reliability in long-term clinical studies of disease-modifying therapies for relapsing-remitting multiple sclerosis: A systematic review

PLOS ONE

Dear Dr. Lucchetta,

Thank you for submitting your manuscript to PLOS ONE. After careful consideration, we feel that it has merit but does not fully meet PLOS ONE’s publication criteria as it currently stands. Therefore, we invite you to submit a revised version of the manuscript that addresses the points raised during the review process.

We would appreciate receiving your revised manuscript by May 02 2020 11:59PM. To enhance the reproducibility of your results, we recommend that if applicable you deposit your laboratory protocols in protocols.io, where a protocol can be assigned its own identifier (DOI) such that it can be cited independently in the future. For instructions see: http://journals.plos.org/plosone/s/submission-guidelines#loc-laboratory-protocols

We look forward to receiving your revised manuscript.

Kind regards,

Aristeidis H. Katsanos, MD, PhD

Academic Editor

PLOS ONE

Reviewers' comments:

Reviewer's Responses to Questions

**Comments to the Author**

1. If the authors have adequately addressed your comments raised in a previous round of review and you feel that this manuscript is now acceptable for publication, you may indicate that here to bypass the “Comments to the Author” section, enter your conflict of interest statement in the “Confidential to Editor” section, and submit your "Accept" recommendation.

Reviewer #1: All comments have been addressed

Reviewer #2: All comments have been addressed

2. Is the manuscript technically sound, and do the data support the conclusions?

Reviewer #1: Yes

Reviewer #2: Yes

3. Has the statistical analysis been performed appropriately and rigorously? 

Reviewer #1: Yes

Reviewer #2: Yes

4. Have the authors made all data underlying the findings in their manuscript fully available?

Reviewer #1: Yes

Reviewer #2: Yes

5. Is the manuscript presented in an intelligible fashion and written in standard English?

Reviewer #1: Yes

Reviewer #2: Yes

6. Review Comments to the Author

Reviewer #1: (No Response)

Reviewer #2: Thank you for your revision. There are some extra minor revisions that you should make. lines 52-54 needs rephrasing, lines 229-232 needs rephrasing, lines 323-325 needs rephrasing. Personalized medicine is not patient's experience in terms of effectiveness and safety. Please see personalized medicine in multiple sclerosis, and the definition of personalized medicine and then rephrase the lines mentioned above. All other corrections were ok

7. PLOS authors have the option to publish the peer review history of their article (what does this mean?). If published, this will include your full peer review and any attached files.

Reviewer #1: No

Reviewer #2: No

---

## [Author Response · Author response to Decision Letter 1]

28 Mar 2020

Dear Aristeidis H. Katsanos,

We would like to thank you and the reviewers for the constructive review and for the suggestions for improving our manuscript for publication in Plos One.

We send below the responses and comments to the editor and reviewers:

Comments to the Author

6. Review Comments to the Author

Reviewer #1: (No Response)

Reviewer #2: Thank you for your revision. There are some extra minor revisions that you should make. lines 52-54 needs rephrasing, lines 229-232 needs rephrasing, lines 323-325 needs rephrasing. Personalized medicine is not patient's experience in terms of effectiveness and safety. Please see personalized medicine in multiple sclerosis, and the definition of personalized medicine and then rephrase the lines mentioned above. All other corrections were ok

Authors: Thanks for the comment. We reviewed the three excerpts (highlighted), considering publications by Giovannoni 2017 and Pellegrini et al. 2019.

---

## [Editor Report · Decision Letter 2]

31 Mar 2020

Reliability in long-term clinical studies of disease-modifying therapies for relapsing-remitting multiple sclerosis: A systematic review

PONE-D-19-31331R2

Dear Dr. Lucchetta,

We are pleased to inform you that your manuscript has been judged scientifically suitable for publication and will be formally accepted for publication once it complies with all outstanding technical requirements.

With kind regards,

Aristeidis H. Katsanos, MD, PhD

Academic Editor

PLOS ONE
---

## [Editor Report · Acceptance letter]

5 Jun 2020

PONE-D-19-31331R2 

Reliability in long-term clinical studies of disease-modifying therapies for relapsing-remitting multiple sclerosis: A systematic review 

Dear Dr. Lucchetta:

I'm pleased to inform you that your manuscript has been deemed suitable for publication in PLOS ONE. Congratulations! Your manuscript is now with our production department. 

Kind regards, 

on behalf of

Dr. Aristeidis H. Katsanos 

Academic Editor

PLOS ONE